# Comparison of long-term patient-reported outcome and quality of life 10 years after severe trauma versus polytrauma

**Adelina Denzel****\*, Annette Keß, Christian Kleber, Georg Osterhoff**

Department of Orthopedics, Trauma and Plastic Surgery University Hospital Leipzig, Germany

\* adelina.denzel@medizin.uni-leipzig.de

## Abstract

### Purpose

Severe trauma and polytrauma are major public health concerns due to their long-term impact on survivors' quality of life (QoL). While advances in trauma care have improved survival rates, long-term functional, psychological, and social outcomes remain inadequately understood. This study evaluates the health-related quality of life (HRQoL), functional deficits, and mortality of trauma patients ten years post-injury, comparing severely injured (SI) and polytrauma (PT) patients.

### Methods

Patients admitted to the shock trauma unit at a Level 1 trauma center between 2010 and 2013 with an Injury Severity Score (ISS) ≥ 9 were identified. Survivors were categorized into SI (ISS ≥ 9 and <16 or isolated injury in one body region) and PT (ISS ≥ 16 and injuries in multiple body regions). HRQoL was assessed using the Polytrauma Outcome (POLO) Chart, including the EuroQol 5D-3L, SF-36, and Trauma Outcome Profile (TOP).

### Results

Ninety-one patients (42 SI, 49 PT) completed follow-up, with an additional 80 confirmed deceased patients. Polytrauma patients had significantly longer hospital and ICU stays, higher rates of mechanical ventilation, and worse functional outcomes (p < 0.001). While PT patients reported more physical impairments and pain (SF-36 Physical Functioning, Physical Role, and Pain dimensions; p < 0.05), no significant differences were observed in psychological or social dimensions. Overall, 60.4% of the cohort reported reduced QoL (EuroQol Index ≤ 0.8), with no significant difference between SI and PT groups. Long-term mortality did not differ between groups (p = 0.6).

**Data availability statement:** Data cannot be shared publicly due to data protection and privacy regulations applicable to patient health information. A pseudonymized version of the dataset is available from the Department of Orthopedics, Trauma and Plastic Surgery, University Hospital Leipzig (contact via ethik@medizin.uni-leipzig.de Trial ID Number 326/23-ek) for researchers who meet the criteria for access to confidential data.

**Funding:** The author(s) received no specific funding for this work.

**Competing interests:** The authors have declared that no competing interests exist.

## Conclusion

Ten years post-trauma, both SI and PT patients experience substantial reductions in QoL, with physical impairments more pronounced in PT patients. However, psychological and social outcomes appear independent of injury severity, suggesting that factors beyond trauma severity influence long-term recovery. These findings highlight important associations relevant to long-term outcomes and underscore the need for further research to better define how persistent pain, functional limitations, and psychosocial factors interact in long-term recovery after severe injury and polytrauma.

## Introduction

Severe trauma and polytrauma represent significant public health concerns due to their high incidence, associated morbidity, and long-term impact on survivors' quality of life [1]. Advances in trauma care have substantially improved with survival rates up to 85–88% [2,3]; however, the long-term consequences of severe injuries, particularly regarding physical, psychological, and social outcomes, remain inadequately understood. While acute care focuses on reducing mortality, there is increasing recognition of the importance of long-term functional recovery and quality of life in trauma survivors. Patients who survive severe trauma often face persistent physical impairments, chronic pain, a reduced participation in the labor market and psychological disorders such as depression, anxiety, and post-traumatic stress disorder, which can significantly affect their reintegration into daily life and overall well-being [2,4,5].

Polytrauma patients, characterized by injuries to multiple body regions and higher Injury Severity Scores (ISS ≥ 16 points) [6], typically require more complex acute care and extended rehabilitation compared to those with isolated severe injuries. Existing literature indicates that polytrauma is associated with prolonged hospital stays, increased need for intensive care, and higher rates of long-term disability [7,8]. However, whether the severity and distribution of injuries directly correlate with long-term health-related quality of life remains a subject of ongoing research. While numerous studies have addressed short- and medium-term outcomes, data on long-term recovery, especially beyond five years post-trauma, are scarce [7]. Understanding the factors influencing long-term outcomes is crucial for optimizing rehabilitation strategies and healthcare resource allocation.

The Polytrauma Outcome (POLO) Chart offers a comprehensive tool for assessing health-related quality of Life (HRQoL) in trauma patients by combining established instruments such as the European Quality of Life Questionnaire (EuroQol 5D-3L), the Short Form Health Survey (SF-36), and the Trauma Outcome Profile (TOP) [7]. These instruments enable a multidimensional evaluation of physical, mental, and social health, providing valuable insights into the recovery trajectory of trauma survivors.

This study aims to bridge the gap in long-term outcome data by presenting ten-year follow-up results of patients with severe injuries and polytrauma treated at a

German Level I trauma center. Specifically, it investigates differences in HRQoL, functional deficits, persistent pain, and mortality of patients with isolated or severe injuries and those with polytrauma.

## Materials and methods

### Patients

This study was designed as a cross-sectional observational study. Patients were selected from our Level 1 trauma center shock trauma unit admission database, based on the following inclusion criteria: Injury Severity Score (ISS) ≥ 9 points, admission between January 2010 and December 2013, age ≥ 18 years, residence in Germany.

Patients who had objected to the use of their data for research purposes were excluded. A total of 503 eligible patients were contacted by mail and were asked to complete a questionnaire at a single time point (Fig 1).

In addition to the data collected from the POLO-Chart, further information was obtained from the hospital's internal prospective trauma database, the clinical information system and the Saxon Population register. This also allowed the inclusion of 80 deceased patients in the study for the endpoint of death.

The patients were divided into two subgroups: severely injured patients (SI) and polytraumatized patients (PT). Patients were classified as severely injured (SI) if they had an ISS of ≥ 9 and < 16 points or an injury confined to only a single body region, aligning with the official definition of serious injuries by the European Commission (AIS ≥ 3) [9,10]. Those with an ISS ≥ 16 points and injuries in more than one body region were categorized as polytrauma (PT) [6,11].

### Polytrauma Outcome (POLO) Chart

The Polytrauma Outcome (POLO) Chart is specifically designed to assess health-related quality of life (HRQoL) in trauma patients. It includes, among others, the European Quality of Life (EuroQol 5D-3L), the Short Form Health Survey (SF-36) and the Trauma Outcome Profile (TOP) [7,12]. All questionnaires are self-administered.

The EuroQol 5D-3L consists of a visual analog scale (VAS) ranging from 0 to 100 representing the respondent's current health status and an index score based on five dimensions: mobility, self-care, usual activities, pain and anxiety/depression [13].

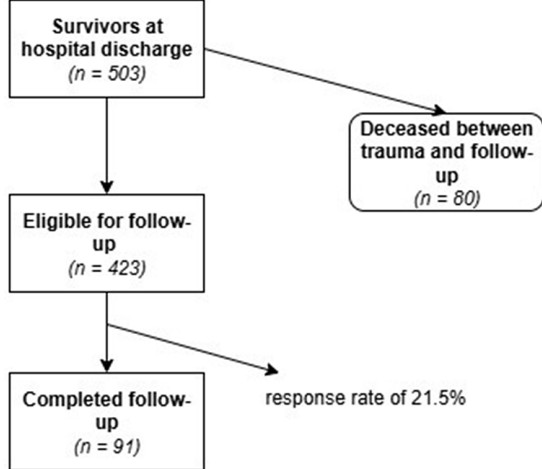

**Fig 1. Flowchart of patient recruitment and selection process.** Flowchart illustrating patient recruitment and selection process, showing the number of survivors at hospital discharge, patients eligible for follow-up, those deceased during the observation period, and the final number of patients who completed follow-up (response rate 21.5%).

The SF-36 is widely used internationally as a disease-independent instrument for assessing HRQoL. It consists of 36 items that are grouped into eight subscales, each ranging from 0 to 100: physical functioning, physical role, pain, general health, vitality, social functioning, emotional role and mental health [14].

The TOP is an instrument specifically designed to assess HRQoL after trauma. It is divided into 10 subscales covering the following dimensions: depression, anxiousness, PTSD, social aspects, pain before and after trauma (for 14 body regions, scale 0–10), physical functioning before and after trauma (for 14 body regions, scale 0–10), daily activities, mental functioning, body image and overall life satisfaction [7].

## Statistical methods

All data were recorded in an Excel database (Microsoft Corp., Washington, DC, USA) and exported to SPSS 29.0 (IBM Inc., Armonk, NY, USA) for statistical analysis. Demographic and clinical data are reported as means with standard deviation (SD) for continuous variables and as counts (n) and percentages for categorical variables. In cases of skewed distributions, the median and interquartile range ($IQR_{25-75}$) are also provided. Analyses were conducted using available-case analysis. To avoid introducing bias by imputing missing values patients with missing data for the respective variable were excluded from the corresponding analysis. Differences between group means were assessed using Student's $t$-test or the Mann-Whitney $U$ test, as appropriate. Categorical variables were analyzed using the chi-squared test. Effect sizes were calculated where applicable, with Cohen's $d$ interpreted as follows: small effect ($\geq 0.2$), medium effect ($\geq 0.5$), and large effect ($\geq 0.8$). Statistical significance was set at $p < 0.05$.

## Ethical considerations

This study was approved by the local ethics committee of University Hospital of Leipzig. (Ethical Committee Trial ID Number 326/23-ek). The Ethics Committee approved the re contact of patients 10 years after injury, based on the assumption of existing consent. All patients contacted by mail received written information about the study and a consent form, which they were required to sign and return in order to participate.

## Results

### Demographics

A total of 171 patients were included in this study. Among them, 80 patients (15.9%) died within 10 years after the trauma. Of the 423 surviving patients, 91 completed the POLO-Chart data (Table 1). This corresponds to a return rate of 21.5%.

The mean age at the initial trauma was 40.8 years (±16.4) in the severely injured group and 44.8 years (±17.8, p = 0.549) in the polytrauma group. In the SI group, 38 patients (90.5%) were male, whereas in the PT group only 32 patients (65.3%, p = 0.016) were male.

In the polytrauma group, the ISS was significantly higher than in the severely injured group (PT 28.6 ± 12.3; SI 11.7 ± 2.2), as anticipated, given how the groups were defined. The average length of hospital stay in the severely injured group was 11.0 days (±8.9) with a median of 8.5 days ($IQR_{25-75}$ 6.0–13.3) compared to 22.4 days (±15.1, p = 0.005) with a median of 16.0 days ($IQR_{25-75}$ 11.0–26.5) in the polytrauma group.

Regarding ICU admissions, 33 patients (78.6%) in the SI group and 48 patients (98.0%, p = 0.003) in the PT group required intensive care treatment. The length of ICU stay (n = 81) differed significantly between groups: patients in the SI group had a mean ICU stay of 2.0 days (±2.1) with a median of 1.0 day ($IQR_{25-75}$ 1.0–3.0), while the PT group had a mean ICU stay of 11.3 days (±12.4, p < 0.001) with a median of 8.0 days ($IQR_{25-75}$ 3.0–16.0). In terms of intubation and mechanical ventilation, 12 patients (28.6%) in the SI group and 27 patients (55.1%, p = 0.01) in the PT group required respiratory support. The duration of mechanical ventilation (n = 39) was notably longer in the PT group, with a mean of 4.5 days (±7.0) compared to 2.0 days (±2.1, p < 0.001) in the SI group.

 

**Table 1. Demographic data, injury severity, clinical course and outcome of severely injured patients and polytrauma patients.**

| | Severely injured | Polytrauma | All | p |
|---|---|---|---|---|
| n (%) | 42 (46.2%) | 49 (53.8%) | 91 (100%) | |
| Age, mean years (SD) | 40.8 (16.4) | 44.8 (17.8) | 43.0 (17.2) | 0.549 |
| Male, n (%) | 38.0 (90.5%) | 32.0 (65.3%) | 70.0 (76.9%) | 0.016 |
| Length of hospital stay, mean d (SD), median (IQR$_{25-75}$) | 11.0 (8.9), 8.5 (6.0-13.3) | 22.4 (15.1), 16.0 (11.0-26.5) | 17.1 (13.8), 12.0 (8.0-24.0) | 0.005 |
| ICU stay, n (%) | 33 (78.6%) | 48 (98.0%) | 81 (89.0%) | 0.003 |
| Length of ICU stay (n=81), mean d (SD), median (IQR$_{25-75}$) | 2.0 (2.1), 1.0 (1.0-3.0) | 11.3 (12.4), 8.0 (3.0-16.0) | 7.0 (10.3), 3.0 (1.0-9.0) | <0.001 |
| Intubation/mech. ventilation, n (%) | 12 (28.6%) | 27 (55.1%) | 39 (42.9%) | 0.011 |
| Duration of mech. ventilation (n=39), mean d (SD), median (IQR$_{25-75}$) | 2.0 (2.1), 0.0 (0.0-0.3) | 4.5 (7.0), 0.5 (0.0-8.0) | 2.6 (5.6), 0.0 (0.0-1.0) | <0.001 |
| ISS, mean (SD) | 11.7 (2.2) | 28.6 (12.3) | 20.8 (12.4) | |
| Glasgow Outcome Scale | | | | <0.001 |
| GOS 3: severe disability, n (%) | 1 (2.4%) | 5 (10.2%) | 6 (6.5%) | |
| GOS 4: moderate disability, n (%) | 5 (11.9%) | 21 (42.9%) | 26 (28.6%) | |
| GOS 5: good recovery, n (%) | 36 (85.7%) | 23 (46.9%) | 59 (64.8%) | |
| | | | | |

Regarding the functional outcome, the Glasgow Outcome Scale (GOS) revealed statistically significant differences between the SI and PT groups (p<0.001). In the SI group, 1 patient (2.4%) had severe disability (GOS 3), 5 patients (11.9%) had moderate disability (GOS 4), and 36 patients (85.7%) showed good recovery (GOS 5). In contrast, in the PT group, the functional outcome was less favorable, with 5 patients (10.2%) experiencing severe disability, 21 patients (42.9%) having moderate disability, and only 23 patients (46.9%) achieving good recovery.

### Pattern of injuries

The distribution of the most frequently injured body regions was similar between the groups, although the overall injury frequency differed significantly in some cases (Table 2).

The most common injuries in the severely injured group were to the extremities/hip at 64.3% compared to 79.6% in the polytrauma group. The second most frequent injury was to the thorax, occurring in 45.2% of the severely injured group and 77.6% of the polytrauma group. head/neck injuries ranked third, affecting 35.7% of patients in the severely injured group and 61.2% in the polytrauma group. However, there were significant differences in the frequency of injuries in the following regions: head/neck (SI 35.7%; PT 61.2%), thorax (SI 45.2%; PT 77.2%), abdomen (SI 23.8%; PT 57.1%;) and external injuries (SI 23.8%; PT 8.2%).

### Patient-reported outcome

The completed patient-reported outcome questionnaires revealed a statistically significant difference between the two groups in Physical Functioning and Physical Role dimensions (SF-36), as well as in the Pain dimension in the SF-36 (Fig 2). In the Pain, Physical Functioning and Physical Role dimensions, the difference was detected with a small effect size (Cohen's d=0.470; 0.497; 0.499) [15].

In the other dimensions, no significant difference was found between the polytrauma patients and severely injured patients ten years after trauma (Fig 2, Fig 3, Fig 4). However, an overall reduced quality of life was observed in both groups, with 60.4% (n=55) of the entire study population reporting a reduced quality of life (EuroQol-Index ≤0.8) and

**Table 2. Localization of initial injured body region of survivors (Abbreviated Injury Scale AIS > 0).**

| Localization of injury | Initial Injuries n (%) | |
|---|---|---|
| | **Severely injured** | **Polytrauma** |
| | 42 (100.0) | 49 (100.0) |
| Head/Neck | 15 (35.7) | 30 (61.2) |
| Face | 7 (16.7) | 12 (24.5) |
| Thorax | 19 (45.2) | 38 (77.6) |
| Abdomen/Pelvic contents | 10 (23.8) | 28 (57.1) |
| Extremities/ Pelvic girdle | 27 (64.3) | 39 (79.6) |
| External | 10 (23.8) | 4 (8.2) |

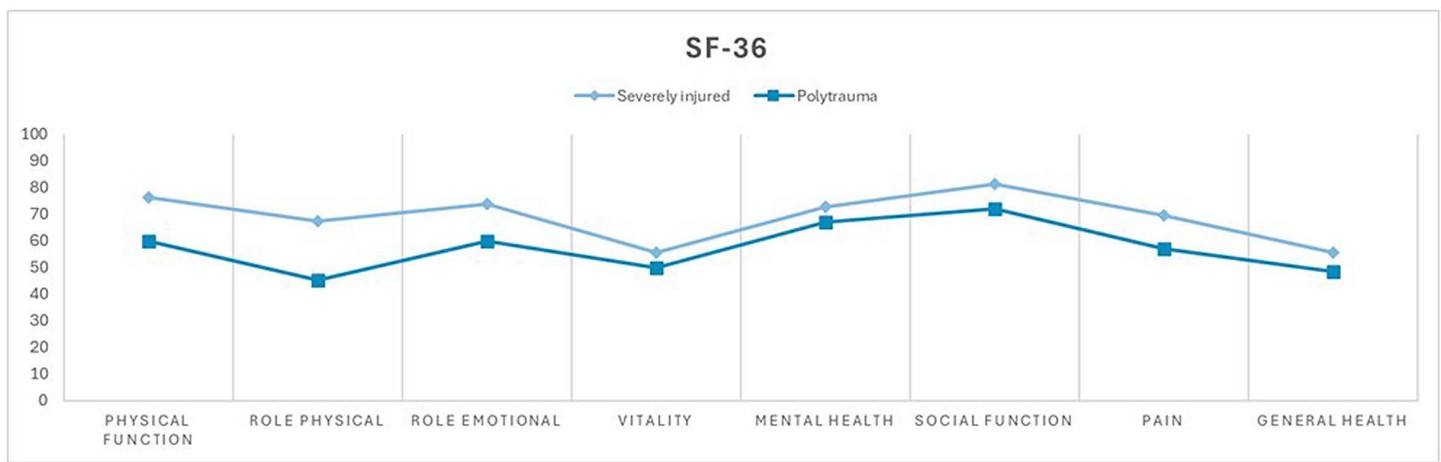

**Fig 2. Quality of life as measured by SF-36.** Comparison of mean values between the severely injured group (ISS ≥ 9 and <16 points or an injury confined to a single body region) and the polytrauma group (ISS ≥ 16 points and injuries in more than one body region). Physical functioning, physical role, pain, general health, vitality, social functioning, emotional role, and mental health on the X-axis, scale values ranging from 0 to 100 on the Y-axis.

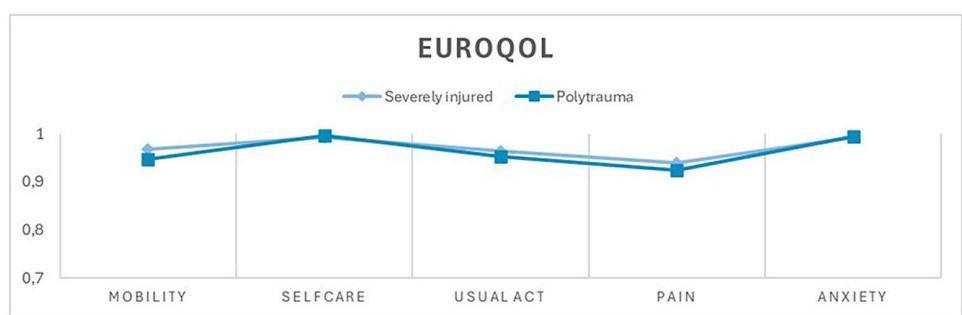

**Fig 3. Current health status as measured by EQ-5D 3L.** Comparison of mean values between the severely injured group (ISS ≥ 9 and <16 points or an injury confined to only a single body region) and the polytrauma group (ISS ≥ 16 points and injuries in more than one body region). Mobility, self-care, usual activities, pain, and anxiety on the X-axis, coefficient values ranging from 0.7 to 1 on the Y-axis.

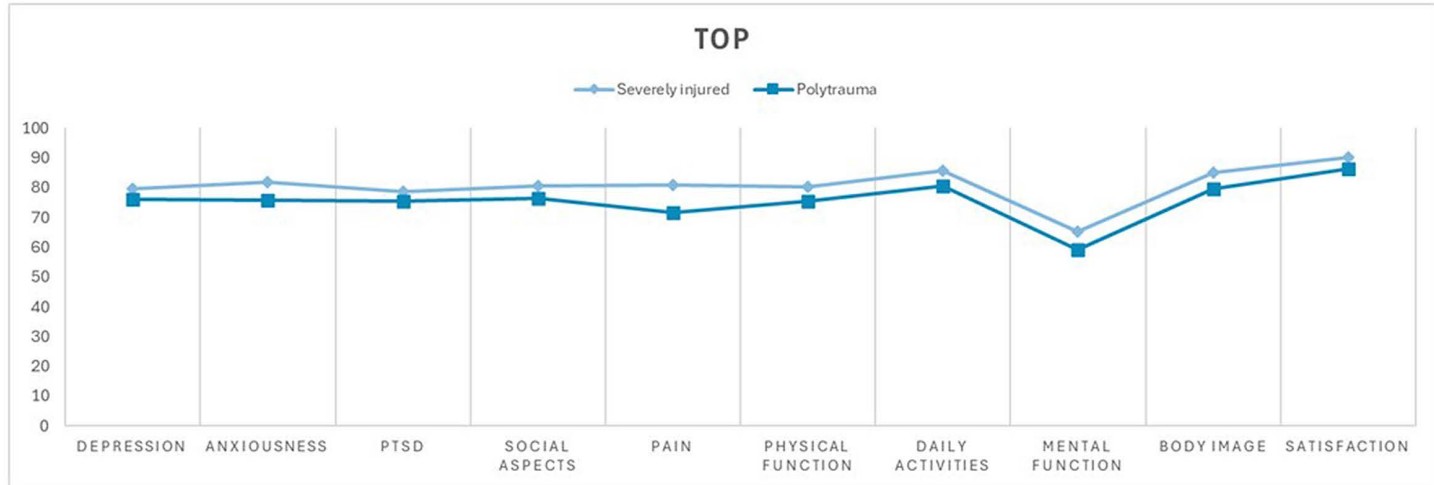

**Fig 4. Trauma-specific impact on quality of life measured by TOP.** Comparison of mean values between the severely injured group (ISS ≥ 9 and <16 points or an injury confined to only a single body region) and the polytrauma group (ISS ≥ 16 points and injuries in more than one body region). Depression, anxiousness, PTSD, social aspects, pain, physical function, daily activities, mental function, body image, and satisfaction on the X-axis. The y-axis shows the corresponding TOP scores ranging from 0 to 100.

20.9% (n = 19) reporting a severely reduced quality of life (EuroQol-Index ≤0.5). The cutoff values were based on normative population data and previous studies using similar cut-offs in trauma populations [16,17]. There was no statistically significant difference between the group of severely injured and polytrauma patients.

## Death

After a period of 10 years, no significant difference in mortality was found between severely injured and polytrauma patients (log-rank test: p = 0.6). The survival rate of the patients that survived the initial trauma and treatment was 50.0% (n = 42) in the severely injured group and 55.1% (n = 49) in the polytrauma group.

## Discussion

In this study, we examined the long-term outcomes of severely injured and polytrauma patients ten years after trauma.

A significant difference between the groups was observed in the dimensions of physical functioning, physical role, and pain in the SF-36 questionnaire. However, no significant differences were found in the other dimensions of the POLO Chart between severely injured and polytrauma patients.

Another notable result of our study is that, in the entire cohort of patients admitted with an ISS ≥ 9 points, 60% exhibited reduced quality of life (QoL) ten years after the injury. Reduced quality of life was defined as a EuroQol index ≤0.8 based on normative population data and previous studies using similar cut-offs in trauma populations [16,17]. Even though 15,9% of our study population died within 10 years after trauma, there was no significant difference in long-term mortality between polytrauma and severely injured patients ten years post-injury.

Our findings align with previous research on long-term trauma outcomes. Kaske et al. reported that two-thirds of severely injured patients exhibited reduced QoL (Euroqol index ≤0.8) two years post-trauma, which is consistent with the 60% observed in our cohort [17]. Other studies also suggest that QoL reduction after trauma occurs independently of injury severity [18]. Similarly, previous research has shown that QoL and mental impairment following trauma do not necessarily correlate with injury severity [2,19,20]. This supports our findings, showing, for the most part, psychological and social dimensions are not primarily influenced by trauma severity alone but rather by other factors.

Simmel et al. investigated long-term outcomes two years post-injury in patients with ISS ≥ 25 points and found that foremost non-trauma-related factors—such as female sex, lower educational background, advanced age, and pre-existing chronic conditions—significantly negatively impact long-term HRQoL [19]. Other studies also suggest that long-term outcomes depend not only on injury severity but also on injury localization [17,18,21,22]. For instance, Pape et al. analyzed data from 637 patients with ISS ≥ 16 points ten years post-injury and found that lower extremity injuries and amputations, two or more articular injuries, or a combination of articular and shaft injuries were strong predictors for poor outcomes. There was no clear correlation found between ISS and QoL in this study as well [21]. Attenberger et al. reported that head and neck injuries are particularly associated with poorer psychological outcomes, while injuries to the extremities and pelvis predominantly affect physical dimensions and body image [22]. However, our data contradicts these findings. While head and neck injuries were significantly more frequent in the polytrauma group, we found no significant differences in the psychological dimension or any impact on the Glasgow Outcome Scale (GOS).

Conversely, although the prevalence of extremity injuries did not differ between the two groups, we observed a significant difference in physical dimensions of the SF-36. Our findings are consistent with Bator et al., whose data suggests that while injury location has no significant impact on physical health, persisting pain after trauma plays a crucial role in influencing physical health and its perception by the patients [10]. In conclusion, multiple studies indicate that ISS alone does not strongly correlate with general QoL. However, trauma-related factors still appear to influence patient outcomes to some degree. It seems especially individual patients' characteristics play a major role in the long-term outcome.

Furthermore, we found no significant difference in long-term mortality between polytrauma and severely injured patients that survived the initial trauma and treatment, ten years after the initial trauma. This is consistent with other studies, such as Kuorikoski et al., which examined ten-year long-term mortality in patients with ISS ≥ 16 points. Their study found an increased mortality rate in the first 30 days after the incident. When differentiating between polytrauma and non-polytrauma patients, long-term survival was primarily influenced by the affected body region, with traumatic brain injuries being associated with significantly higher mortality rates [23]. Even though there is a significant difference in head/neck injuries between the SI and PT groups in our study, this does not translate into differences in survival rates after ten years. One possible explanation is survivor bias, patients with severe consequences from head/neck injuries may die earlier, leading to a convergence of mortality rates over time. This aligns with other studies suggesting that while ISS is a useful parameter for the initial management of trauma patients, its predictive value concerning mortality rate diminishes in the long-term course [24,25].

Despite these insights, our study has several limitations. The response rate was typically low, with only 91 out of 423 patients (21.5%) completing follow-up, alongside 80 out of 503 confirmed deceased patients (15.9%). This low response rate introduces a potential selection bias, as patients with poorer outcomes may have been less likely to participate. The very long follow-up period of ten years represents a major challenge for patient tracking, as life circumstances change over time, which may further contribute to loss to follow-up and potentially affect the results. Comparable drop-out rates have been reported in previous studies with similarly long follow-up periods [26,27]. Also, a predominance of male patients was observed in our cohort, particularly among the severely injured group. This finding is consistent with well-established epidemiological patterns in trauma populations, where men are more frequently affected by high-energy trauma mechanisms such as traffic accidents, occupational injuries, and other risk-related activities. Similar sex distributions have been reported in national and international trauma registries, including the German TraumaRegister DGU®, which shows that approximately 70% of severely injured patients are male [28,29]. The study thus reflects a sex distribution that is typical for severely injured and polytraumatized populations. Another important consideration in follow-up studies is the potential influence of non-response. However, due to the lack of consent, baseline data for non-responders could not be collected. Consequently, potential systematic differences between responders and non-responders could not be assessed, which may introduce bias and should be considered when interpreting the long-term outcome findings. In addition to that given by the observational nature of the study, residual confounding by factors not captured in the dataset cannot be entirely

excluded. Therefore, between-group comparisons should be interpreted with caution, and the results should be regarded as exploratory rather than causal. Nevertheless, our study has notable strengths. Access to the Saxon Population Registry allowed us to accurately determine the number of deceased patients, reducing uncertainty in long-term survival estimates and providing an accurate representation of the mortality rate. The ten-year follow-up period also provides valuable insights into trauma outcomes, particularly which role the severity of the injury plays.

### Clinical relevance

Our findings highlight the necessity of long-term care strategies that go beyond injury severity alone. Despite significant differences in physical impairment, polytrauma and severely injured patients show similar decreased overall quality of life (QoL) outcomes ten years post-injury. This suggests that rehabilitation programs should follow a more holistic approach, independent of the Injury Severity Score (ISS), with a greater focus on individual patient characteristics. Factors such as social support, pre-existing conditions, and psychological resilience may be crucial in optimizing long-term recovery. Additionally, persistent pain and functional deficits remain major challenges for patients after trauma. These impairments are most commonly observed in the extremities and spine region, underlining the need for targeted rehabilitation strategies. A better understanding of non-injury-related determinants of long-term outcomes could help tailor interventions to improve patient-centered care.

Future studies should focus on identifying key predictors of long-term QoL beyond injury severity. Specifically, the role of psychosocial factors, mental health status, and access to rehabilitation services warrants further investigation. Additionally, longitudinal studies with larger cohorts and higher response rates are needed to validate our findings and minimize selection bias. Exploring the impact of specific injury patterns, such as traumatic brain injury or lower extremity fractures, on long-term functional outcomes could provide valuable insights for individualized treatment strategies. Furthermore, research should examine whether early intervention programs, including psychological support and multidisciplinary rehabilitation, can mitigate the long-term burden of trauma.

### Conclusion

This study demonstrates that, ten years after trauma, both severely injured and polytrauma patients experience significantly reduced quality of life, with 16% mortality but no major differences in long-term mortality between the groups. While polytrauma patients exhibit greater physical impairments and functional deficits, psychological and social dimensions appear to be influenced by factors beyond injury severity alone. These findings highlight important associations relevant to long-term outcomes and underscore the need for further research to better define how persistent pain, functional limitations, and psychosocial factors interact in long-term recovery after severe injury and polytrauma.

### Author contributions

**Conceptualization:** Adelina Denzel, Georg Osterhoff.

**Data curation:** Adelina Denzel, Annette Keß, Christian Kleber, Georg Osterhoff.

**Formal analysis:** Adelina Denzel, Georg Osterhoff.

**Project administration:** Georg Osterhoff.

**Supervision:** Christian Kleber, Georg Osterhoff.

**Visualization:** Adelina Denzel.

**Writing – original draft:** Adelina Denzel, Georg Osterhoff.

**Writing – review & editing:** Adelina Denzel, Annette Keß, Christian Kleber, Georg Osterhoff.

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
