## [Decision Letter · Decision Letter 0]

5 Jan 2026

Dear Dr. Denzel,

Thank you for submitting your manuscript to PLOS ONE. After careful consideration, we feel that it has merit but does not fully meet PLOS ONE’s publication criteria as it currently stands. Therefore, we invite you to submit a revised version of the manuscript that addresses the points raised during the review process.

We look forward to receiving your revised manuscript.

Kind regards,

Pasyodun Koralage Buddhika Mahesh

Academic Editor

PLOS One

2. In the online submission form you indicate that your data is not available for proprietary reasons and have provided a contact point for accessing this data. Please note that your current contact point is a co-author on this manuscript. According to our Data Policy, the contact point must not be an author on the manuscript and must be an institutional contact, ideally not an individual. Please revise your data statement to a non-author institutional point of contact, such as a data access or ethics committee, and send this to us via return email. Please also include contact information for the third party organization, and please include the full citation of where the data can be found.

3. Please remove your figures from within your manuscript file, leaving only the individual TIFF/EPS image files, uploaded separately. These will be automatically included in the reviewers’ PDF.

Reviewers' comments:

Reviewer's Responses to Questions

**Comments to the Author**

1. Is the manuscript technically sound, and do the data support the conclusions?

Reviewer #1: No

Reviewer #2: Partly

Reviewer #3: Yes

2. Has the statistical analysis been performed appropriately and rigorously?

Reviewer #1: Yes

Reviewer #2: Yes

Reviewer #3: Yes

3. Have the authors made all data underlying the findings in their manuscript fully available?

Reviewer #1: No

Reviewer #2: Yes

Reviewer #3: Yes

4. Is the manuscript presented in an intelligible fashion and written in standard English?

Reviewer #1: Yes

Reviewer #2: Yes

Reviewer #3: Yes

Reviewer #1: As the authors have correctly mentioned, the low response rate is affecting the quality of the study as well as the accuracy of the inferences made based on the results. with this it is difficult to draw conclusions based on this sample.

Reviewer #2: Results

• Table 2: The percentage for chest for polytrauma – is shown as "61.2%". Similar to “Face” . It’s likely a copy paste error , please correct it (should be 24.5% for PT based on 12/49)

• In the demographic data(Table 1), males show a significant percentage among those who were injured compared to females. It is better if this can be discussed meaningfully in the discussion.

Death – The statement says ‘The survival rate was "50.0% (n=42) SI vs 55.1% (n=49) in the PT group". But these are the responders (42+49=91), not all patients. Calculating survival rate this way leads to major bias.

Conclusion –

• From the initial sample of 503 participants, after accounting for the 80 deaths we have 423 survivors. However, the responders are only 91 (21%). Drawing recommendations and conclusions from this limited subset likely excludes the worst outcomes, introducing survivorship bias, a limitation that is not acknowledged in the conclusion

• "emphasize the need for individualised rehabilitation" however this study design can't support causal claims

Reviewer #3: This manuscript addresses an important and under-researched topic by examining long-term patient-reported outcomes and mortality ten years after severe trauma and polytrauma. The extended follow-up period and use of validated outcome measures, together with registry-based ascertainment of mortality, represent notable strengths.

Several methodological and reporting issues, however, require clarification to improve transparency and interpretability. The response rate among surviving patients is low (21.5%), raising substantial concerns regarding selection and non-response bias. The manuscript would benefit from clearer reporting that allows readers to assess whether responders differ systematically from non-responders, as well as a more detailed discussion of the potential direction and magnitude of any resulting bias. A baseline comparison of responders versus non-responders among surviving patients (e.g., age, sex, ISS, ICU stay, mechanical ventilation, major injury regions, if available) should be provided.

The study design is not explicitly stated. Clear identification of the study design is important for transparency and alignment with reporting standards for observational studies.

Analyses are largely unadjusted despite baseline differences between groups and the observational nature of the data. This limits interpretation of between-group comparisons, and the implications of residual confounding should be more clearly addressed.

Reporting of participant flow, non-responder characteristics, and missing data handling is incomplete. In particular, the manuscript does not describe how missing questionnaire items or incomplete scales were managed during analysis.

Further clarification is also needed regarding the derivation of EQ-5D-3L index scores and the rationale for the thresholds used to define reduced and severely reduced quality of life (≤0.8 and ≤0.5). Without this information, interpretation and reproducibility are limited.

Several minor issues should also be addressed, including improving consistency in terminology, correcting typographical and formatting errors, clarifying follow-up contact procedures, and correcting figure numbering to follow standard conventions (the use of “Figure 0” is non-standard).

I have no concerns regarding research ethics, consent procedures, or potential dual publication based on the information provided. Addressing the above issues would substantially improve the clarity and robustness of the manuscript.

what does this mean?. If published, this will include your full peer review and any attached files.). If published, this will include your full peer review and any attached files.

**Do you want your identity to be public for this peer review?** For information about this choice, including consent withdrawal, please see our For information about this choice, including consent withdrawal, please see our Privacy Policy .

Reviewer #1: No

Reviewer #2: No

Reviewer #3: **Yes:** I.O.K.K.NanayakkaraI.O.K.K.Nanayakkara

---

## [Author Response · Author response to Decision Letter 1]

22 Jan 2026

We thank all reviewers for their valuable and important comments. We have carefully addressed these suggestions in the revised manuscript and have provided detailed responses in a separate response-to-reviewers document submitted with the revision.

---

## [Decision Letter · Decision Letter 1]

11 Mar 2026

Comparison of long-term patient-reported outcome and quality of life 10 years after severe trauma versus polytrauma

PONE-D-25-51161R1

Dear Dr. Denzel,

We’re pleased to inform you that your manuscript has been judged scientifically suitable for publication and will be formally accepted for publication once it meets all outstanding technical requirements.

Kind regards,

Pasyodun Koralage Buddhika Mahesh

Academic Editor

PLOS One

Additional Editor Comments (optional):

Reviewers' comments:

Reviewer's Responses to Questions

**Comments to the Author**

Reviewer #2: All comments have been addressed

Reviewer #3: All comments have been addressed

2. Is the manuscript technically sound, and do the data support the conclusions?

Reviewer #2: Yes

Reviewer #3: Yes

3. Has the statistical analysis been performed appropriately and rigorously?

Reviewer #2: Yes

Reviewer #3: Yes

4. Have the authors made all data underlying the findings in their manuscript fully available?

Reviewer #2: Yes

Reviewer #3: Yes

5. Is the manuscript presented in an intelligible fashion and written in standard English?

Reviewer #2: Yes

Reviewer #3: Yes

Reviewer #2: The authors have undertaken careful and thoughtful revisions to the manuscript, addressing the previously raised concerns comprehensively. I sincerely appreciate the time, attention, and diligence invested in refining the work. These revisions have significantly enhanced the clarity of reporting, strengthened the methodological transparency, and improved the overall scientific rigor of the manuscript

Reviewer #3: • The authors have provided a substantially improved revised manuscript, addressing the majority of the concerns raised in the initial review.

• The study design is now clearly stated as a cross-sectional observational study, improving transparency and alignment with STROBE reporting recommendations.

• Participant flow is clearly described and illustrated, including the number of eligible patients, deceased patients, and the final response rate at ten-year follow-up.

• The low response rate (21.5%) and the potential for selection and non-response bias are now explicitly acknowledged and appropriately discussed as key limitations.

• The handling of missing data has been clarified, with analyses conducted using available-case analysis and without imputation.

• The derivation of EQ-5D-3L index scores and the justification for the selected cut-off values defining reduced and severely reduced quality of life are now clearly explained and supported by relevant references.

• Figure numbering and formatting issues noted in the previous review have been appropriately corrected.

• Although multivariable analyses and a baseline comparison between responders and non-responders would have strengthened the study, the authors have provided a reasonable justification for not performing these analyses, citing limited sample size, data protection constraints, and risk of model overfitting.

• The Discussion section has been strengthened by framing the results as exploratory and associative, rather than causal, which is appropriate given the observational study design.

• Overall, the remaining limitations are inherent to long-term follow-up studies of this nature and are transparently acknowledged.

• In summary, the authors have adequately addressed the major and minor concerns raised in the initial review, and the manuscript is now methodologically sound, clearly reported.

what does this mean?. If published, this will include your full peer review and any attached files.). If published, this will include your full peer review and any attached files.

**Do you want your identity to be public for this peer review?** For information about this choice, including consent withdrawal, please see our For information about this choice, including consent withdrawal, please see our Privacy Policy .

Reviewer #2: No

Reviewer #3: **Yes:** I.O.K.K.NanayakkaraI.O.K.K.Nanayakkara

---

## [Editor Report · Acceptance letter]

PONE-D-25-51161R1

PLOS One

Dear Dr. Denzel,

I'm pleased to inform you that your manuscript has been deemed suitable for publication in PLOS One. Congratulations! Your manuscript is now being handed over to our production team.

Kind regards,

on behalf of

Dr. Pasyodun Koralage Buddhika Mahesh

Academic Editor

PLOS One